# Stiff Extracellular Matrix Promotes Invasive Behaviors of Trophoblast Cells

**DOI:** 10.3390/bioengineering10030384

**Published:** 2023-03-21

**Authors:** Jialing Cao, Hangyu Li, Hongyan Tang, Xuenan Gu, Yan Wang, Dongshi Guan, Jing Du, Yubo Fan

**Affiliations:** 1Key Laboratory for Biomechanics and Mechanobiology of the Ministry of Education, Institute of Nanotechnology for Single Cell Analysis, Beijing Advanced Innovation Center for Biomedical Engineering, School of Biological Science and Medical Engineering, Beihang University, Beijing 100083, China; 2Sino-French Engineer School, Beihang University, Beijing 100083, China; 3State Key Laboratory of Nonlinear Mechanics, Institute of Mechanics, Chinese Academy of Sciences, Beijing 100190, China; 4School of Engineering Science, University of Chinese Academy of Sciences, Beijing 100049, China; 5Department of Obstetrics and Gynecology, Peking University Third Hospital, Beijing 100191, China

**Keywords:** embryo implantation, human choriocarcinoma cell, extracellular matrix, stiffness, durotaxis

## Abstract

The effect of extracellular matrix (ECM) stiffness on embryonic trophoblast cells invasion during mammalian embryo implantation remains largely unknown. In this study, we investigated the effects of ECM stiffness on various aspects of human trophoblast cell behaviors during cell–ECM interactions. The mechanical microenvironment of the uterus was simulated by fabricating polyacrylamide (PA) hydrogels with different levels of stiffness. The human choriocarcinoma (JAR) cell lineage was used as the trophoblast model. We found that the spreading area of JAR cells, the formation of focal adhesions, and the polymerization of the F-actin cytoskeleton were all facilitated with increased ECM stiffness. Significantly, JAR cells also exhibited durotactic behavior on ECM with a gradient stiffness. Meanwhile, stiffness of the ECM affects the invasion of multicellular JAR spheroids. These results demonstrated that human trophoblast cells are mechanically sensitive, while the mechanical properties of the uterine microenvironment could play an important role in the implantation process.

## 1. Introduction

Embryo implantation is a critical feature of mammalian pregnancy and requires a series of interactions between the embryo and the uterus, which can be divided into three different steps: apposition, attachment, and invasion [1]. In brief, after hatching from the zona pellucida, the spheroid of cells floats freely and finds the proper implantation site (apposition), and the trophoblasts firmly attach to the uterine wall (adhesion). The trophoblasts then differentiate into cytotrophoblasts and syncytiotrophoblasts under tight regulation, thereby invading the endometrium (invasion) and leading to compromised placental development and pregnancy complications [2].

Recent research on embryo implantation primarily focused on biochemical aspects, including some molecular mechanisms and related signaling pathways. During the early stages of implantation, there are many molecular mediators coordinated by ovarian steroid hormones involved in the initial maternal-fetal communication and interaction. These mediators include adhesion molecules, cytokines, growth factors, and lipids [3,4]. Meanwhile, using human endometrium Ishikawa and PL95-2 cells, Ming Yu et al. found that N-glycosylation of the endometrium is necessary to maintain the receptive functions of the uterus [5]. 

Several studies recently uncovered the role of mechanical forces before and during human embryo implantation [6,7,8]. The mechanical properties of the human endometrium are complex, and spatiotemporal changes exist during embryo implantation [8,9]. Mechanical indentation was used to demonstrate that mechanical properties significantly differ between anatomical locations in both nonpregnant and pregnant uterine tissue [10]. For instance, in pregnant tissue, Young’s modulus of fundus tissue is higher than that of the posterior and anterior tissue. The stiffness of decidual basalis (implantation site) is much higher than the decidua parietalis, nonpregnant endometrium, and placenta (~10^3^ Pa vs. ~10^2^ Pa) [11]. When the uterus is in a pathological state or there is a scar in the endometrium, the stiffness will increase [12,13]. For example, the shear modulus in a pathological state is about 7 kPa (calculated elastic modulus: 18.2 kPa), and the shear modulus in an extreme pathological state is about 17.4 kPa (calculated elastic modulus: 45.2 kPa) [6,13]. Endometrium extracellular matrix (ECM) stiffness is lower during the secretory phase than during the proliferative phase (shear modulus: 3.34 kPa vs. 1.97 kPa; calculate elastic modulus: 8.68 kPa vs. 5.12 kPa) [14].

Recent work demonstrated that the implantation process may be regulated by mechanical factors. Firm adhesion between trophoblast-type cells and endometrium epithelial cells was only observed when the trophoblast-functionalized tip indented the apical surface of the epithelial cell [15], indicating the role of mechanical forces during the maternal-fetal interaction [7]. Meanwhile, recent work studied the mechanobiological regulation of placental trophoblast fusion through ECM stiffness [6]. Using a stiffness-tunable hydrogel culture system, Ma et al. suggested that stiffer ECM promotes the spreading and fusion of trophoblast cells [6]. Wong et al. demonstrated that thicker ECM promotes the self-assembly of 3D trophoblast cells spheroids [16]. These might suggest an effect of ECM stiffness on the invasion of trophoblast cells, but they do not directly demonstrate how substrate stiffness regulates the invasion behaviors of trophoblast cells. Therefore, it is important to reveal the effect of ECM stiffness on trophoblast cell adhesion, migration, and invasion of multicellular spheroids.

Polyacrylamide hydrogels are widely used as a biocompatible material with easy fabrication and adjustable stiffness to study the effect of substrate stiffness on cell behaviors [17,18,19]. In this study, we generated collagen-coated polyacrylamide (PA) hydrogels with different stiffness gradients to mimic the microenvironment of the uterus and investigate the role of ECM stiffness in trophoblast cell morphology, migration, and invasion. We chose the human choriocarcinoma (JAR) cell lineage, which is derived from human choriocarcinoma and is the most widely-used cytotrophoblast-like cell model [20,21,22,23,24,25,26,27]. Our results showed that as the simulated ECM stiffness increased, the spreading area of JAR cells gradually promoted, and the number of focal adhesions increased, while the cytoskeleton became more robust and filamentous. We also found that individual trophoblast cells exhibited durotaxis behavior, and simulated ECM stiffness enhanced the invasion capacity of trophoblast spheroids. This study complements the effect of mechanical forces on the invasion behaviors of trophoblast cells, revealing the important role of ECM stiffness in embryonic implantation, while being able to provide some references for the study of diseases related to implantation.

## 2. Materials and Methods

### 2.1. General Cell Culture

The JAR human choriocarcinoma cell line (TCHu156 ATCC) was maintained in RPMI 1640 medium (Hyclone) at 37 °C under 5% CO_2_ in incubator. The culture media was supplemented with 10% (*v*/*v*) fetal bovine serum (FBS, SERANA), 100 units/mL penicillin, and 100 units/mL streptomycin.

### 2.2. JAR Spheroid Formation

For the spheroid formation assay, we seeded 100 μL of JAR cells at a density of 1 × 10^5^ cells/mL in each well of an ultra-low attachment 96-well round bottom plate (Corning, NY, USA). Cells were incubated overnight at 37 °C in a 5% CO_2_ incubator. During this incubation, JAR cells formed spheroids of 50–200 μm in diameter via natural aggregation. The aggregate diameter of the spheroids was measured by CellSens standard software (Olympus, Tokyo, Japan).

### 2.3. Fabrication of Polyacrylamide Hydrogels with Stiffness Gradients 

A polymer solution containing 20% (*v*/*v*) acrylamide monomer (Bio-Rad) and 0.4% (*v*/*v*) N,N′-methylenebisacrylamide cross-linker (Bio-Rad) was prepared in 1× Dulbecco’s phosphate-buffered saline (PBS, Life Technologies, Gaithersburg, MD, USA) without Mg^2+^ and Ca^2+^. An amount of 1 mL aliquots were degassed for 5 min, and 4 µL of 10% (*w*/*v*) ammonium persulfate (APS) (Sigma-Aldrich) was added and mixed for 0.5 s; then, 1.2 µL of N,N,N′,N′-tetramethylethylenediamine (TEMED) (Bio-Rad) was quickly added and mixed for 1 s. An amount of 120 µL of this solution was transferred into a glass mold, and a functionalized glass coverslip was placed over the solution at an angle so that no air remained under the coverslip. The solution was allowed to polymerize for 30 min at room temperature, after which the coverslip was removed from the mold. The gels were rinsed and stored in 1× PBS without Mg^2+^ and Ca^2+^ for subsequent use. For cell adhesion, Sulfosuccinimidyl 6-(4′-azido-2′-nitrophenylamino) hexanoate (sulfo-SANPAH, Thermo Fisher Scientific, Waltham, MA, USA) was applied to the PA gel surface for 15 min under 365 nm UV light (2 times). Then, the gel was coated with collagen I (100 µg/mL) overnight at 4 °C. Before the cell seeding, PA gels were washed 3 times with PBS. PA hydrogels with this structure have thickness gradient and apparent stiffness gradient [28]. 

To measure the thickness of the gel, PA gel was stained with Coomassie brilliant blue G-250 for about 10 min. Using confocal microscope (Leica TCS SP8 X, Germany, Wetzlar, Germany), we obtained a series of Z-stack images and reconstructed to measure the thickness of the PA gel. 

An atomic force microscope (AFM, MFP-3D, Asylum Research Inc., Santa Barbara, CA, USA) with the colloidal probe cantilever was used for the apparent Young’s modulus measurement. The cantilever used in the experiments was a non-tip cantilever (NSC36, MikroMasch, Watsonville, CA, USA) with a spring constant of 2 N/m. The colloidal probe was assembled, as described previously, by adhering a glass sphere of diameter d = 26.3 μm to the front end of the cantilever [29]. To effectively reduce the adhesion between the probe and the PA gel, the surface of the colloidal probe was coated with a thin layer of PLL-g-PEG (SuSoS AG.) [30]. The AFM measurements were carried out in contact mode with 1~8 µm indentation depths. The measured force–distance curves were recorded and fitted with Hertz model $F=\frac{4}{3\left(1-\upsilon^{2}\right)}{\mathrm{ER}}^{0.5}\delta^{1.5}$ to calculate the reduced Young’s modulus E [31], where F is the measured force, R is the probe radius, δ is the indentation distance, and υ=0.33 is the Poisson ratio. Prior to each force measurement, the spring constant of the colloidal probe is calibrated in situ using the thermal power spectral density method [29].

The topography of PA gel surface was visualized using scanning electron microscopy (SEM, Quanta 200 FEG). PA gels were flash frozen and were then lyophilized overnight. A layer of platinum (Pt) was deposited on the gel surface using a turbomolecular pump coater (Q150T, Quorum Technologies, Lewis, UK) prior to observation to enhance the electrical conductivity of the gel. 

In each independent experiment, the PA gel was fabricated under the same condition. The size, thickness, and partitioning of the PA were controlled consistently. The reproducibility of the gel fabrication was verified.

### 2.4. Cell Immunofluorescence

JAR cells in hydrogel substrates were fixed in 4% (*w*/*v*) paraformaldehyde in PBS for 15 min at room temperature. The hydrogel was washed twice with PBS, permeabilized in 0.1% (*v*/*v*) triton X-100 in PBS for 15 min, and washed twice more with PBS. JAR cells were blocked in 2.5% (*v*/*v*) goat serum in PBS for 2 h at room temperature to prevent non-specific binding. Then, cells were incubated with anti-vinculin antibody (1:200, Abcam, GR3270283-14, overnight, 4 °C) in goat serum. For secondary staining, cells were washed twice with PBS and incubated with goat anti-rabbit IgH H&L (488 nm) antibody (Abcam, 1:500, GR320844-3, 3 h, room temperature). Directly stained cells were incubated with 1:200 DAPI and phalloidin (546 nm) in goat serum solution (2 h, room temperature) and thoroughly washed with PBS. Confocal microscope (Leica TCS SP8 X, Germany, Wetzlar, Germany) was used for acquiring images.

### 2.5. Image Analysis and Figure Preparation

Unless otherwise stated, images were adjusted and analyzed using the Fiji distribution of ImageJ. R was used to generate all graphs and perform all statistical analyses. Figures were made using Photoshop Creative Cloud and PowerPoint.

### 2.6. Live Cell Imaging, Cell Tracking, and Migration Analyses

Cells plated on PA hydrogels were incubated overnight in RPMI 1640 with 10% FBS. For cell tracking, 1 × 10^6^ cells were seeded in the confocal dishes (BioFroxx, BDD-12-35) and incubated overnight. Cells were monitored using an automated live-cell imager (Leica, Germany) with a 10× dry objective and maintained at 37 °C in a 5% CO_2_ environment during imaging. Phase-contrast images were captured every 15 min for 14 h. Imaris’ cell tracking module was used to track individual cells and obtain (x, y) coordinates to calculate displacement, track length, and cell velocity (track_length/time). Migrating cells were identified as those that migrated beyond a circular area 2 times the diameter of the cell over 14 h of imaging. The maximum displacement was calculated as the maximum change in the Euclidean distance of a particular cell throughout the imaging process.

### 2.7. F-Actin Skeletonization

F-actin skeletonization was performed using a steerable filter with MATLAB and ImageJ [32]. In brief, multiple-scale steerable filtering was used to enhance the curvilinear features, and centerlines of curvilinear features were extracted. Next, the filament fragments were clustered into high and low confidence and the fragments were connected with a graph matching. A network of F-actin was obtained after the reconstruction and each filament was represented by an ordered chain of pixels and a local filament orientation. ImageJ was used to determine the length of detected filaments (Plugins > NeuronJ) and enhance filament thickness for visualization. 

### 2.8. Focal Adhesion Area Measurement

Focal adhesion area was measured using ImageJ software [33]. Briefly, the original image of immunofluorescence was firstly subtracted the local background by applying SUBSTRACT BACKGROUND. Then, the local contrast of the image was enhanced by adjusting Contrast Limit Histogram Equalization. Mathematical exponential and BRIGHTNESS & CONTRAST tool were used for further minimizing the background. Next, we used the LOD3D plugin to filter the image and ran the THRESHOLD command to convert the image to a binary image. Finally, ANALYSE PARTICLES command was executed to scan the binary image and find the edge of focal adhesions.

### 2.9. Particle Image Velocimetry (PIV) Measurement

PIV analysis was performed using a custom algorithm based on MATLAB’s PIVlab software package (Matlab2020a, PIVlab2.36). We used live cell image sequences of JAR cells to analyze the direction and magnitude of cell movement. To avoid the influence of the background movement on the calculated results, the calculated velocity field was subtracted from the average velocity. For the velocity vectors arrows, each pixel of length represents 0.05 μm/min.

### 2.10. Durotaxis Assays

Cells were seeded on PA hydrogels with stiffness gradients. The cells were mounted and maintained on the microscope as described above. After live cell imaging and cell tracking, the direction of cell movement was calculated based on the results of the PIV analysis (fractional velocities in the x and y directions). 

### 2.11. Spheroid Spreading Assay

Multicellular spheroids were generated as described above. Diameters of JAR spheroids were similar to those of human embryos at the periods of implantation (5–6 days after fertilization), with an average of 150 ± 15 μm [34,35]. By observation under a stereomicroscope (Olympus, Japan), each spheroid at the bottom of the well was carefully aspirated with a disposable pipette tip and transferred into a small dish (LABSELECT 12111). Spheroids with proper size were gently collected and evenly dispersed into six-well plates containing hydrogel-coated coverslips. Spheroids were then incubated for 24 h for attachment and spreading before imaging with 10× or 20× objectives. To quantify the degree of dispersion, images were firstly converted into 8-bit image and then thresholded using ImageJ (Image > Adjust > Threshold) to outline the periphery of the aggregate, and the spreading area (total area–spheroid area) of the 24 h image was divided by the area of the spheroid, which was considered the spreading ratio.

### 2.12. Statistical Analysis

All statistical analyses were performed using R software (version 4.1.1). Results were expressed as the mean ± standard deviation (SD). Every experiment was repeated three times (*n* = 3). After confirming that the data were normally distributed and homogeneous in variance using the Shapiro–Wilk significance test as well as the Bartlett test, Student’s *t*-test was used for analysis. For all comparisons, *p* < 0.05 was considered statistically significant.

## 3. Results

### 3.1. Simulated ECM Stiffness Regulates Trophoblast Cell Morphology and Spreading Area

Numerous studies demonstrated that cell spreading and focal adhesion maturation are positively correlated with ECM rigidity in various cells [36,37,38,39,40,41,42]. However, this correlation was not investigated in cytotrophoblasts. Therefore, we fabricated PA hydrogels (Appendix A), examined the surface morphology of the gels by SEM (Appendix A), and measured the apparent elastic modulus by AFM (Appendix A), which represent varying ECM stiffnesses independently of topographical and compositional cues [28]. We cultured JAR cells on simulated ECM with stiffness gradients ranging from 10 kPa to 100 kPa (Figure 1A,B). We then divided the hydrogel into three regions according to their stiffness: a stiff region (46.7 ± 25 kPa), an intermediate region (14.6 ± 8.0 kPa), and a soft region (6.9 ± 0.5 kPa).

After 24 h incubation, the cells were observed by phase-contrast microscope (Figure 1C). Cells growing on simulated ECM with different stiffness showed significant changes in cell morphology and spreading area. On the stiffer simulated ECM, JAR cells were polygonal with a larger cell spreading area, while the spreading area of the cells tended to decrease as stiffness decreased, and the cell morphology gradually became round (Figure 1D). These results demonstrate that JAR cell morphology and spreading area are regulated by simulated ECM stiffness.

### 3.2. F-Actin Organization and Focal Adhesion Formation Are Affected by Simulated ECM Stiffness in JAR Cells

Since cell spreading is regulated by the cytoskeleton and focal adhesion complex [43], we measured the assembly of F-actin and the focal adhesion in JAR cells on simulated ECM with different stiffnesses (Figure 2A). The images revealed significant differences in actin organization and focal adhesion formation between three simulated ECM regions with different stiffnesses. The focal adhesion area indicated by the staining of vinculin, which was measured by ImageJ software, increased as simulated ECM stiffness increased (Figure 2B). To quantify the F-actin cytoskeleton structure differences, we applied a steerable filter to extract F-actin bundles [32] (Figure 2C). The results demonstrated that JAR cells in the stiff simulated ECM region had longer and more robust stress fibers compared to other simulated ECM regions (Figure 2D). This demonstrates that stiff simulated ECM enhanced the F-actin organization and focal adhesion assembly of JAR cells.

### 3.3. Stiff Simulated ECM Enhances JAR Cell Motility

Several studies demonstrated that cell migration is regulated by ECM stiffness [44,45,46,47,48]. To investigate cell migration behavior on ECM with different stiffness, we used a time-lapse microscope to visualize the motility of JAR cells. To examine the relationship between ECM stiffness and JAR cell migration, we used particle image velocity (PIV) to analyze the movement of JAR cells on simulated ECM with different stiffnesses (Figure 3A). Meanwhile, we selected JAR cells in different simulated ECM regions (stiff, intermediate, and soft) and tracked their migration for 5 h (Figure 3B). The migration distance (track length) as well as the migration velocity (track_length/time) of JAR cells on the inter and stiff regions were significantly increased compared to JAR cells on the soft region (Figure 3C,D). The displacement of JAR cells significantly increased as simulated ECM stiffness increased (Figure 3E). These results indicate that stiff simulated ECM increases cell motility in JAR cells.

### 3.4. JAR Cells Exhibit Durotaxis

Spatial changes in ECM stiffness were shown to induce migration toward increased stiffness in numerous cell types both in vitro and in vivo [49]. This process, which is a key regulator of cell migration and invasion, is called durotaxis [50,51,52]. Although durotaxis was observed in many cell types [53,54,55,56], few studies described durotaxis in the context of human trophoblast cells. Therefore, we performed tracing for JAR cells grown on simulated ECM with a large stiffness gradient for 9.5 h (Figure 4A). By analyzing these trajectories, we found that most cells tended to migrate toward the stiffer simulated ECM region (Figure 4B,D). PIV analysis that was performed on two adjacent frames also demonstrated that the migratory direction of most cells coincided with the positive direction of the stiffness gradient (Figure 4C). These results demonstrate that JAR cells exhibit durotaxis.

### 3.5. Stiff Simulated ECM Enhances the Adhesion and Invasion of Multicellular JAR Spheroids

During the complex biophysical process of embryo implantation, trophoblasts contribute to successful implantation via attachment and invasion. Numerous studies demonstrated many similarities between embryo implantation and tumor progression [57,58,59,60]. Components that are crucial to tumor cell migration and invasion are shared by the human trophoblast, including the involvement of the extracellular matrix (ECM), proteases (including serine proteases, cathepsins, and matrix-metalloproteinases), and cell-surface receptors (integrins) [59]. F-actin remodeling regulated by fascin plays a critical role in both cancer metastasis and trophoblast migration and invasion [60,61].

As such, we inferred that if the migration of individual JAR cells is affected by the mechanical forces of their microenvironment, the invasion behaviors of multicellular JAR spheroids could also be regulated by ECM stiffness. During the first step of the embryo implantation process, the blastocyst, a spheroid, establishes adhesion to the endometrium. Wong et al. demonstrated that ECM stiffness regulates the self-assembling of 3D placental trophoblast spheroids [16], but few studies directly demonstrated how ECM stiffness affects adhesion or spreading of 3D human trophoblast spheroids.

Based on this, we referred to tumor research methods to study the effect of ECM stiffness on invasion of multicellular JAR spheroid [62]. Multicellular JAR spheroids were formed in ultra-low attachment 96-well plates, as described in Methods, and were then seeded onto PA hydrogels with different stiffness and allowed to attach for 24 h (Figure 5A). JAR spheroids on stiff and intermediate-stiff simulated ECM had larger adhesion areas and showed a higher degree of invasion compared to spheroids on soft simulated ECM (Figure 5B–D). No significant difference was found in the degree of spheroid invasion on the stiff and intermediate-stiff simulated ECM. This demonstrated that the invasion of multicellular JAR cell spheroids is regulated by simulated ECM stiffness.

## 4. Discussion

Although ECM stiffness was demonstrated to be a key regulator of several developmental processes, the importance of extracellular mechanics for embryo implantation, especially for embryo attachment, was not established. This work identified trophoblasts as mechano-responding malignant tumor-like cells. Using PA hydrogels that mimic Young’s modulus values of the human endometrium, we demonstrated that stiffer substrate enhances various cellular processes closely related to trophoblast adhesion and invasion, including JAR cell morphology, migration, contractility, and multicellular spheroid disaggregation. Our results demonstrate that adhesion and invasion of trophoblasts could be regulated by the mechanical properties (e.g., stiffness) of the endometrium. Altogether, these results suggest that mechanobiological properties may regulate the adhesion and invasion of human embryo during the process of implantation, and that the stiffness of the endometrium may affect the selection of embryo implantation sites and the subsequent invasion process.

Scar pregnancy (CSP) occurs when an embryo implants on a scar in the uterus, and the incidence of CSP increases with the number of previous cesarean deliveries [63]. The exact pathogenesis of CSP is not known, but the mechanical properties of the uterine scar significantly differ from other sites. The stiffness of the uterine scar appears to be significantly increased compared to the intact myometrium, as measured by ultrasound elastography [12], which could also affect embryo implantation. In addition, the stiffness of the endometrium in the pathological state is significantly higher than that of the normal state [6,13]. Our experiments demonstrated that in the context of normal and diseased human endometrium tissue characteristics (difference in stiffness), the mechanobiological regulation of trophoblast migration and attachment likely plays a critical role in implantation.

More broadly, this work demonstrated that trophoblast migration and adhesion are mechanically sensitive, which highlights the importance of strategies using extracellular tissue engineering to better understand and develop treatments for diseases related to implantation. This knowledge of mechanically mediated mechanisms for migration and adhesion can be further leveraged to create better technologies to increase the success rate of in vitro fertilization (IVF). In addition, the identification of this process will help to identify new regulatory mechanisms of embryonic adhesion and provide new ideas for the development of therapeutic strategies related to pregnancy.

Considering the difficulty in obtaining pure, primary, first-trimester human trophoblast cells, we used human choriocarcinoma (JAR) cell lineage instead of primary trophoblast cell line in this study. However, compared with primary trophoblast cells, choriocarcinoma cell lines have different transcriptomic profiles, are malignant and contain an abnormal number of chromosomes, which is unfavorable for studying the uniquely invasive extravillous trophoblast (EVT) cell behavior [64]. Therefore, in the subsequent study of embryo invasion, we will choose human embryonic stem cell (hESCs) or adult progenitor cells to derive trophoblast organoids. Endometrial epithelial cells are also involved in the embryo implantation process [65,66]. Under normal conditions, the trophoblast cells interact with endometrial epithelial cells to achieve maternal–fetal adhesion. The stiffness of the ECM could also affect the function of endometrial epithelial cells and the expression of related proteins. Therefore, to better simulate the in vivo environment, it is necessary to consider the mechanism of interaction between these two cells under the regulation of mechanical properties. Furthermore, since mechanical stiffness affects the invasion behaviors of trophoblast cells, the mechanobiological regulation of trophoblast migration and adhesion is most likely to be related to integrin-related signaling pathways. The integrin protein mediates the adhesion between cells and ECM. After being affected by mechanical forces, integrin binds to its ligands and mediates FAK, PI3K, AKT/PKB, and other signaling pathways that regulate cell proliferation, migration, and epithelial-mesenchymal transition [67,68]. By upregulating integrin-β1, the invasion of human trophoblasts can be promoted [69]. We also examined the expression of integrin-β1 of JAR cells on different regions and found no significant differences either between soft and inter or between inter and stiff, with stiff being slightly higher than soft (Appendix A), and the expression of other subunits of integrin did not change significantly with substrate stiffness [16]. In addition, several studies demonstrated the significant role of Rho protein in the process of implantation [26,70,71]. For example, Rho GTPase, most likely RhoA, regulates the adhesion of human trophoblasts to uterine epithelial RL95-2 cells [26].RhoA can also regulate trophoblast migration through cytoskeleton reorganization [70]. By interfering with related proteins such as integrin and Rho, we can further study the precise molecular mechanism of this phenomenon, which is also the content of our follow-up research.

## Figures and Tables

**Figure 1 bioengineering-10-00384-f001:**
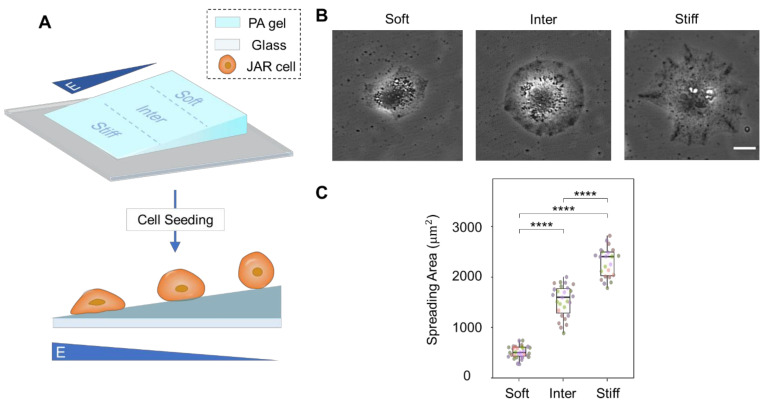
Simulated ECM stiffness regulates trophoblast cell morphology and spreading area. (**A**) (**Top**): schematic diagram of PA gel with stiffness gradient. The Young’s modulus of the PA gel is 1 kPa, and its apparent Young’s modulus gradually increases with the decrease in the gel thickness. According to the change of the apparent Young’s modulus, the surface of the PA gel is evenly divided into three regions, namely Stiff, Intermediate, and Soft. (**Bottom**): schematic representation of JAR cells cultured on different regions. E indicates the apparent Young’s modulus. (**B**) Phase contrast images of JAR cells cultures on different regions scale bar: 20 μm. The yellow dashed line indicates the boundary of cells. (**C**) Measured average cell spreading area of JAR cells cultured on different regions. **** *p* < 1 × 10^−6^, *n* = 31, 27, 23 for Soft, Inter and Stiff, respectively. Data reported as mean ± standard deviation for *N* = 3 independent experiments. Each scatter indicates each cell being measured, and each color indicates an independent experiment.

**Figure 2 bioengineering-10-00384-f002:**
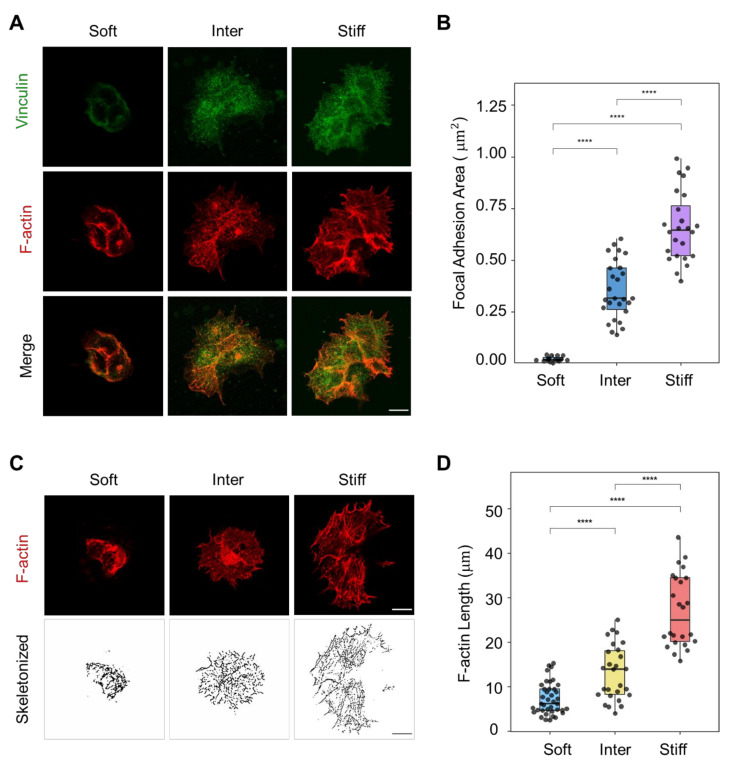
F-actin organization and focal adhesion formation are affected by simulated ECM stiffness in JAR cells. (**A**) Immunofluorescence staining of JAR cells cultured on different regions (red: F-actin; green: vinculin, scale bar: 20 μm). (**B**) Measured focal adhesion area of JAR cells cultured on different regions. **** 1 × 10^−6^, *n* = 18, 27, 24 for Soft, Inter, and Stiff, respectively. Data reported as mean ± standard deviation for *N* = 3 independent experiments. Each scatter indicates each focal adhesion being measured. (**C**) Skeletonization of F-actin in JAR cells cultured on different regions (scale bar: 20 μm). (**D**) Measured cytoskeleton length of JAR cells cultured on different regions. **** 1 × 10^−6^, *n* = 35, 27, 24 for Soft, Inter, and Stiff, respectively. Data reported as mean ± standard deviation for *N* = 3 independent experiments. Each scatter indicated each F-actin filament being measured.

**Figure 3 bioengineering-10-00384-f003:**
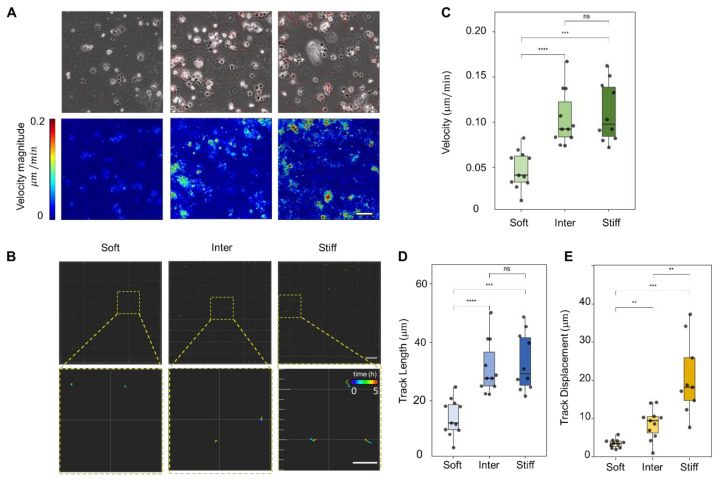
Stiff simulated ECM enhances JAR cell motility (**A**) (**Top**): Vectors of JAR cell migration. (**Bottom**): heatmap of velocity magnitude on different regions of simulated ECM (scale bar: 50 μm, color bar: 0~0.2 μm/min). (**B**) Tracking of JAR cells cultured on different regions (scale bar: 100 μm, time bar: 5 h). Each scatter indicated each cell being analyzed. (**C**,**D**) Velocity and track length of JAR cell migration. **** *p* < 1 × 10^−6^, *** *p* < 0.001, *n* = 11, 11, 9 for Soft, Inter, and Stiff, respectively. Data reported as mean ± standard deviation for *N* = 3 independent experiments. Each scatter indicated each cell being analyzed. (**E**) Track displacement of JAR cells cultured on different regions. *** *p* < 0.001, ** *p* < 0.01, *n* = 11, 11, 9 for Soft, Inter, and Stiff, respectively. Data reported as mean ± standard deviation for *N* = 3 independent experiments.

**Figure 4 bioengineering-10-00384-f004:**
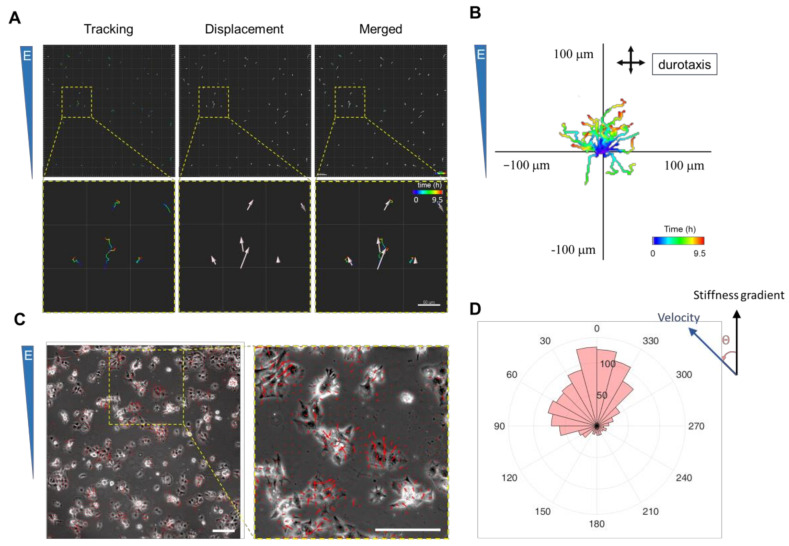
JAR cells exhibit durotaxis. (**A**) Tracking of JAR cells cultured on stiff region (scale bar: 50 μm, time bar: 9.5 h). E indicates the apparent Young’s modulus. Arrows indicated the displacement of each cell, (**B**) Representative JAR cell migration plots on stiff region over 9.5 h. The total cell number *n* = 56, number of independent experiments *N* = 3. (**C**) Vector map of JAR cell migration on stiff region (scale bar: 150 μm). Arrows indicated the vector of velocity. (**D**) Rose diagram of cell migration direction, which displays the angular between migration and stiffness gradient and the frequency of each class. The total cell number *n* = 56, number of independent experiments *N* = 3.

**Figure 5 bioengineering-10-00384-f005:**
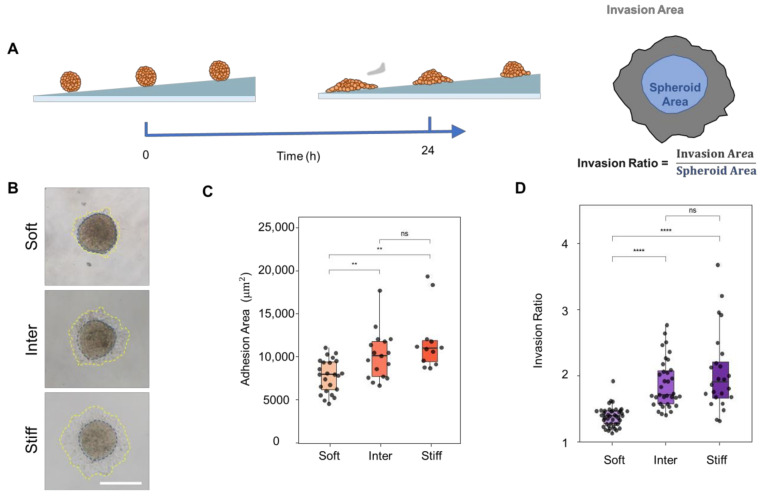
Stiff simulated ECM enhances the adhesion and invasion of multicellular JAR spheroids. (**A**) (**Left**): schematic diagram of JAR spheroid invasion assay. (**Right**): schematic diagram of the calculation of the invasion ratio. (**B**) Image of JAR spheroid invasion taken by an inverted microscope (scale bar: 200 μm). (**C**) Calculated adhesion area of JAR spheroids on different regions. ** *p* < 0.01, *n* = 24, 17, 12 for Soft, Inter, and Stiff, respectively. Data reported as mean ± standard deviation for *N* = 3 independent experiments. Each scatter indicated each spheroid being measured. (**D**) Calculated invasion ratio of JAR spheroids on different regions. **** *p* < 1 × 10^−6^, *n* = 43, 35, 25 for Soft, Inter, and Stiff, respectively. Data reported as mean ± standard deviation for *N* = 3 independent experiments. Each scatter indicated each spheroid being analyzed.

## Data Availability

The data presented in this study are available on request from the corresponding author. The data are not publicly available due to privacy.

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
