# Peer review of "Stiff Extracellular Matrix Promotes Invasive Behaviors of Trophoblast Cells"

_bioengineering, 2023, doi:10.3390/bioengineering10030384_

Round 1

Reviewer 1 Report (New Reviewer)

General comments:

In this manuscript, the authors generated PAA hydrogels with different levels of stiffness to study their effects on a choriocarcinoma cell line (JAR). The choice of PAA hydrogel to study the influence of stiffness on cell spreading, migration, and even differentiation is appropriate due to its finely controlled and tunable stiffness. However, the novelty of the study and its impact is low. Past studies have already reported the effect of substrate/ECM stiffness on some of these aforementioned cell properties. For example, Ma Z and colleagues showed that different levels of stiffness affect trophoblast spreading (Sci Rep. 2020 Apr 3;10(1):5837.) using a similar PAA approach. Although a major caveat from those studies and the present one is the fact that they used as cytotrophoblast models (CTBs) immortalized cell lines (such as BeWo and JAR cells) that, do not recapitulate most of the biological characteristics of CTBs (Stem Cell Reports. 2016 Feb 9; 6(2): 257–272). Currently, we have better models such as the Okae human Trophoblast stem cells (hTSC; Cell Stem Cell. 2018 Jan 4;22(1):50-63.e6.) that display four of the characteristics that define first-trimester CTBs. An even more accurate model to study the effects of substrate stiffness on embryo implantation could have been the use of expanded potential-potential stem cells, although I understand the technical difficulties and ethical approvals required for these types of studies.

Comments on the manuscript:

COMMENT: The title statement is quite broad, could the authors change it for a title that accurately states what the study is about?

COMMENT: In the abstract, the authors did not mention the use of JAR cells. This information should be included.

It is not clearly stated in the introduction the reasons for performing this study. What is the gap in knowledge?

Materials and methods comments:

In line 88, lack of information on the manufacturer origin of the ultra-low attachment 96-well plated.

In line 100, could the authors specify at what temperature was the gel allowed to polymerize for 30 minutes?

In line 106, specify the microscope used to obtain Z-stack images.

In line 107, can the authors spell out AFM? Maybe add in parenthesis “Atomic Force Microscope”.

In the section Cell immunofluorescence, can the authors state the microscope used for acquiring images?

In lines 136 and 138, RPMI is lacking before 1640.

In line 137, what type of dishes were used?

For F-actin skeletonization, did the authors use an ImageJ plugin to determine the length of the filaments? If so, could state which one?

In line 171, can the authors refer to which study shows the average size of embryos at the time of implantation? Are they referring to human embryos? Specify, please.

In line 172, can the authors explain how they collected the spheroids for further analysis?

How focal adhesion was measured?

Results comments:

Is the level of stiffness of the PAA matrix (3 different levels) consistently reproduced in the 3 independent experiments shown in Figure 1? Can the 3 independent experiments be colour coded in the scattered dot plots?

In section 3.2 a reference for "Since cell spreading is regulated by the cytoskeleton and focal adhesion complex “ is required. The authors can also include this reference (Pelham RJ, Wang YL. PNAS. 1997;94:13661–13665) from one of the first studies done in 1997 reporting the importance of the mechanical properties of the substrate on cell migration and focal adhesion.

Since cells have been shown to sense levels of stiffening (in other cell models, mostly in the cancer field) through integrins, does this increase in focal adhesion translate into the upregulation of certain integrins? It is well known that differential integrin expression on CTBs plays important roles, not only in migration and invasion but also in trophoblast differentiation (ITGA1/ITGA2/ITGA5/ITGA6). An assay showing alterations of integrin expression/levels due to different substrate stiffness would be recommended. For this section, it would have been interesting to see how the substrate stiffness not only increases the number of focal adhesion but how this translates into the recruitment of focal adhesion molecules and activation of intracellular signalling. Did the authors considered examine RhoA since it has been reported that its activity is increased in stiff substrates (Journal of Cell Biology. 2003;163:583–595., Journal of Cell Biology. 2001;153:1175–1186. The Journal of Clinical Endocrinology & Metabolism, Volume 87, Issue 12, 2002, Pages 5808–5816, https://doi.org/10.1210/jc.2002-020376).

How do the authors explain that the distance of migration is significantly higher in stiff vs intermediate, but velocity showed no significant difference between those two groups? Here it would be informative to have the different experiments colour-labelled to see the correlation between velocity and distance.

In section 3.5 the authors show that a stiff substrate enhances the invasion of spheroids compared to a soft ECM. However, they did not mention that even at an intermediate substrate stiffness invasion is enhanced at the same level, showing no differences between the intermediate and stiff substrate. This should be mentioned in this result section, otherwise, the state made by the authors is misleading.

“Few studies have described durotaxis in the 259 contexts of human trophoblast cells.“ please add the reference for these studies.

This sentence from line 286 “to our knowledge, no study has directly demonstrated how ECM stiffness affects adhesion or spreading in spheroid human trophoblast cells. “ https://journals.plos.org/plosone/article?id=10.1371/journal.pone.0199632 on supplemental figure 1 they show that the thickness of their gels (they performed 3D cell spheroids) is inversely proportional to their stiffness. In agreement with the results shown by the authors in Figure 1, M.K Wong et al showed in their study from 2018 that thinner (stiff matrix) lead to increased spread areas. They also showed that stiffness sensing led to the expression of genes associated with migration and invasion in a thinner (i.e., stiffer) matrix Figure 7 of the aforementioned paper.

Author Response

Reviewer 2 Report (New Reviewer)

Peer-Review bioengineering – 2215199

The research manuscript entitled “The mechanical microenvironment regulates the invasive behaviours of trophoblast cells” from Jialing Cao et al. is a very interesting study on the effects of uterus ECM stiffness on the morphology, migration and invasion of trophoblast cells. The manuscript fits well within the scope of the journal Bioengineering (ISSN 2306-5354), especially in the Special Issue “Biomechanics-Based Motion Analysis”. However, the manuscript has some issues that must be addressed before being considered for publication.

Major Issues:

1.     The novelty of the study should be addressed, for example, in the final part of the Introduction section.

2.     (Page 2, subsection 2.2.): The authors should describe the methods used to determine the aggregate diameter of the spheroids.

3.     The authors should provide a gross image of the hydrogel used (for example as supplementary material).

4.     The authors need to perform SEM analysis of the hydrogel to assess its microstructure, which is an important factor for cell adhesion and migration.

5.     The manuscript should include a subsection in the “Materials and Methods” describing the Statistical analysis of the data and the statistical tests used.

6.     (Page 5, Figure 1B): The authors should provide clear and higher quality phase contrast images of the cells.

7.     Please clarify: The “n” refers to the number of cells considered in each independent experiment (“N”). Why the differences in the number of cells/spheroids analyzed? For example in Figure 5C, the number of spheroids considered in the “soft” condition is half of the ones analyzed on the “stiff” condition.

8.     (Page 8, Figure 4D caption): Please explain in more detail the meaning of the “Rose diagram” (in the figure caption or in the respective Materials and Methods subsection).

9.     The authors should report values/range or values available in the literature for the human endometrium ECM stiffness.

10.  The authors should discuss in more detail why PAA is a good candidate to model the uterus ECM stiffness. Why not use a hydrogel based native ECM components such as collagen or glycosaminoglycans?

11.  (Page 10, 4. Discussion section, lines 327-330): The authors claim that they performed experiments considering diseased human endometrium tissue characteristics. The authors need to justify this statement in more detail (or rectify it).

Other Issues:

1.     The authors should define JAR and ECM (or use the full form) in the keywords section (page 1, line 26).

2.     (Page 1, line 45): The authors need to define the meaning of JAR in its first appearance in the manuscript text.

3.     (Page 2, subsection 2.1., line 83): The word “incubator” is missing.

4.     (Page 3, subsection 2.3., line 97): For consistency with the rest of the manuscript, the authors should use numbers.

5.     (Page 3, subsection 2.4. line 128): The authors should report the incubation temperature.

6.     (Page 3, subsection 2.6. lines 136 and 138): The word “RPMI” is missing.

7.     (Figure 1 caption, lines 198 and 200): Please standardize the use of “PAA” or “PA” throughout the whole manuscript.

Author Response

Reviewer 3 Report (New Reviewer)

the manuscript is interesting and generally well written. To my opinion, it can be accepted in the present form.

Author Response

Thank you very much for your patience and comments.

Reviewer 4 Report (New Reviewer)

In the paper: The mechanical microenvironment regulates the invasive behaviors of trophoblast cells the authors  investigate the effects of ECM stiffness on various aspects of human trophoblast cell behaviors during cell ECM-interactions. The study show that human trophoblast cells are mechanically sensitive, while the mechanical properties of the uterine microenvironment could play an important role in the implantation process. In my opinion, the work is very interesting, innovative; the study is well conducted and very well edited. Therefore I accept the manuscript in the current form.

Author Response

Thank you very much for your patience and comments.

Round 2

Reviewer 1 Report (New Reviewer)

The authors have addressed all of my major concerns and significantly improved the manuscript. I still have some minor comments:

On page 2, line 77, the authors cited the work of Michael K. Wong, but I do not think that it is necessary to add the full name of the authors; only their family name "Wong et al." Same on page 11, line 359.

On page 5 line 204, I believe there is a typo on "BROGTHNESS & CONTRAST" instead of "BRIGHTNESS & CONTRAST."

In the last paragraph of the discussion, the authors mention no significant differences in integrin-b1 expression between soft and inter or between inter and stiff. They did, however, show a significant increase in MFI (median fluorescence intensity) of integrin-b1 between soft and stiff (p = 0.045) in their response letter. This data can be incorporated into the manuscript in supplementary figures.

Author Response

Reviewer 2 Report (New Reviewer)

Dear authors,

The research manuscript entitled "Stiff extracellular matrix promotes invasive behaviors of trophoblast cells" was considerably improved by the authors during this revision process. All the reviewers' comments/suggestions were properly addressed.

Author Response

Thank you very much for your patience and comments!

This manuscript is a resubmission of an earlier submission. The following is a list of the peer review reports and author responses from that submission.

Round 1

Reviewer 1 Report

The authors replied on each comment sincerely and the replies were appropriate. The quality of papers submitted for consideration includes enough reader's interest and scientific quality. The given paper satisfies requirements for publication of this journal.

Reviewer 2 Report

Authors answered to all my requests and I congratulate for their study